materials science

stainless steel, anti-corrosion, layered materials, transition metal dichalcogenides (TMD), MoS$_2$, boron nitride

**Author for correspondence:**
Shakir Bin Mujib
e-mail: sbmujib@ksu.edu

$^\dagger$Equal contribution.

# Assessing corrosion resistance of two-dimensional nanomaterial-based coatings on stainless steel substrates

Shakir Bin Mujib$^\dagger$, Santanu Mukherjee$^\dagger$, Zhongkan Ren and Gurpreet Singh

Department of Mechanical and Nuclear Engineering, Kansas State University, Manhattan, KS 66506, USA

 SBM, 0000-0002-6699-420X; SM, 0000-0002-1190-1117; ZR, 0000-0003-4280-2899 

Two-dimensional (2D) materials have elicited considerable interest in the past decade due to a diverse array of novel properties ranging from high surface to mass ratios, a wide range of band gaps (insulating boron nitride (BN) to semiconducting transition metal dichalcogenides), high mechanical strength and chemical stability. Given the superior chemo-thermo-mechanical properties, 2D materials may provide transformative solution to a familiar yet persistent problem of significant socio-economic burden: the corrosion of stainless steel (SS). With this broader perspective, we investigate corrosion resistance properties of SS-coated with 2D nanomaterials; molybdenum disulfide (MoS$_2$), BN, bulk graphite in 3.5 wt% aqueous NaCl solution. The nanosheets were prepared by a novel liquid phase exfoliation technique and the coatings were made by a paint brush to achieve uniformity. Open circuit potential (OCP) and potentiodynamic plots indicate the best corrosion resistance is provided by the MoS$_2$ coatings. Superior performance of the coating is attributed to low electronic conductivity, large flake size and uniform coverage of SS substrate, which probably impeded the corrosive ions from the solution from diffusing through the coating.

## 1. Introduction

### 1.1. Background and problem identification

Recent advancements in the field of two-dimensional (2D) materials have led to various applications such as nanoelectronics, electrochemical energy storage, photonics, chemical and biological sensing [1–3]. Besides, single-layer and few-layer nanosheets have

**Figure 1.** Impact of corrosion on aspects of modern life.

high chemical stability and the potential to prevent the passage of ions. For example, graphene has demonstrated impermeability to even helium ions [4,5]. Layered crystals also can be exfoliated into atomically thin sheets via simple strategies such as mechanical and chemical exfoliation, thereby allowing production at gram or kilogram levels [6,7].

With this context in mind, 2D materials can be applied to remedy the problem of corrosion, the process by which metal atoms on a surface oxidize and reduce the metal needed to sustain a load [8,9]. Further damage to the structure occurs when cracks form and propagate, potentially leading to catastrophic failure. Stainless steels (SS) (approx. 10.5–28% Cr by weight) are common alloys that also experience corrosion [10]. Corrosion predominantly occurs at damage-prone points such as heat-affected zones and weld points [11–13].

The cost to remedy corrosive effects is approximately 3% of the GDP of industrialized nations [14]. Because the use of SS has been increasing at a rate of 5%, especially in the last two decades, corrosion is a significant problem that must be resolved [14,15]. Figure 1 illustrates the considerable impact of corrosion on various areas of modern life.

## 1.2. The current status of corrosion-resistant coatings and rationale

Coating an exposed surface of SS, or any structural metallic material is the most prevalent technique for corrosion prevention [16,17]. However, metallic coatings (laser-remelted Al) have demonstrated poor performance when subjected to large wear loads [18], and inorganic pigments have concerning toxicity [19]. Nanoparticle metal oxide coatings have also been considered for corrosion protection, but thin-film metal oxide coating techniques such as atomic layer deposition (ALD) are expensive and energy-intensive. In addition, insufficient hydrophilicity of binder composites can severely impair the efficacy and performance of organic coatings [20]. Stojanovic *et al*. [21] studied the effect of polyester and epoxy-polyester based coatings in aggressively corrosive media and found that coating surface texture significantly impacts coating adhesion as well as coating performance. Wang *et al*. [22] found a superhydrophobic polyvinylidene fluoride (PVDF)/carbon nanofibre (CNF) coating to be highly stable under acidic and alkaline conditions for a period of up to 15 days with a water contact angle (WCA) of 164°. Similarly, Radwan *et al*. [23] investigated a nanocomposite coating of PVDF–ZnO obtained by electrospinning and determined that the hydrophobicity of the coating (contact angle approx. 155°) provided enhanced corrosion resistance. Taghavikish *et al*. [24] compiled a detailed analysis and review of the various types of coatings and their anti-corrosion behaviour.

Several works in the literature focus on the application of bulk 2D materials for anti-corrosion coatings with mixed results [25,26]. Although graphene has shown excellent corrosion resistance as a coating for metallic substrates, it is cathodic with respect to almost all metals (and steels), leading to galvanic corrosion [27,28]. To overcome the galvanic corrosion, very uniform and thin graphene layers must be coated on the steel substrates, the application of which may become a manufacturing challenge [29].

Stand-alone and composite $MoS_2$ has been a focus of research interest due to its anti-corrosion properties. ArunKumar et al. [30] investigated the use of doped (Fe, Co and Ni) $MoS_2$ nanosheets for corrosion protection of steel, and they determined that Fe, Co and Ni increase the anti-corrosion properties of the $MoS_2$ coatings, especially for the $MoS_2$–Fe system. Cardinal et al. [31] studied a Ni–W–$MoS_2$ composite material and its frictional properties as a coating material, demonstrating that increasing $MoS_2$ content leads to a porous sponge-like structure with reduced friction coefficient. Li et al. [32] studied boron nitride (BN) nanosheets grown by a chemical vapour deposition technique with the aim of metallic corrosion prevention; results indicated that BN coatings prevent oxidation by increasing the open circuit potential (OCP). However, only a limited amount of literature comprehensively compares various exfoliated 2D materials as anti-corrosion coatings.

The primary objective of this work was to use exfoliated sheets of bulk 2D materials, fabricated by a novel and facile technique, to create anti-corrosion coatings and provide a comparative study. Because fabricating sheets of 2D materials may not be necessary or sufficient for improving corrosion-resistant properties, this research investigated how coating uniformity affected corrosion resistance and sought to obtain a standardized set of results to ascertain the overall corrosion resistance of these materials.

# 2. Material and methods

The raw materials used in this study were commercial SS-304, sodium chloride (NaCl, Sigma Aldrich™, 99.9%), N-methyl pyrrolidone (NMP) and PVDF (Sigma Aldrich™, 99%), graphite flakes (Sigma Aldrich™, 99%), chlorosulfonic acid ($ClSO_3H$, Sigma Aldrich™, 99%) and isopropanol (Sigma Aldrich™, 99.7%). Bulk $MoS_2$ (Sigma Aldrich™, 99.9%) and BN (Sigma Aldrich™, 99.9%) powders were further processed (exfoliated) to obtain the individual layers.

## 2.1. Fabrication of exfoliated layers from bulk two-dimensional materials

### 2.1.1. Exfoliation of molybdenum disulfide

Exfoliation of powder $MoS_2$ (particle size less than 2 µm) was performed by a novel, scalable acid-assisted technique developed previously by this research group and successfully applied for electrochemical energy storage applications [33].

$MoS_2$ powder was carefully weighed out (1 g), and chlorosulfonic acid was added by individual drops (approx. 1 drop every 5 s) to $MoS_2$ so that the concentration of $MoS_2$ in chlorosulfonic acid was approximately 10 mg ml$^{-1}$. Then the solution/dispersion was sonicated for approximately 5 min, after which the solution was allowed to sit for about 1 h to allow the non-exfoliated $MoS_2$ to settle at the bottom. The supernatant remained at the top of the solution because it contained significantly lighter exfoliated $MoS_2$ flakes. The exfoliated sheets were carefully extracted from the top of the solution with a pipette and mixed with deionized (DI) water (approx. 1.0 l) inside a glove box. Finally, the solution was dried in a conventional oven to obtain dried superacid-assisted-exfoliated $MoS_2$. Previous research confirmed that this acid-based exfoliation process successfully provides intact sheets of exfoliated $MoS_2$ [33,34]. Figure 2 is a schematic representation of this chlorosulfonic acid-assisted exfoliation process for obtaining $MoS_2$ flakes.

### 2.1.2. Exfoliation of boron nitride

One gram of powder BN (particle size less than  µm) was carefully weighed out and dispersed in isopropanol to achieve a concentration of approximately 50 mg ml$^{-1}$. The mixture was then sonicated for about 30 min. Similar to the previous case, the sonicated product was allowed to settle for around 1 h, and then the supernatant (exfoliated BN sheets) was collected with a pipette. The supernatant was dried in a conventional oven to obtain the exfoliated BN nanosheet powder. BN, as exfoliated by this technique, has previously shown promise for electrochemical energy storage [35].

## 2.2. Preparation of slurry and coatings

Superacid-assisted-exfoliated $MoS_2$, propanol-assisted-exfoliated BN and powder graphite (flake size 100 mesh) provided the coatings and are henceforth called the active materials. Approximately 30 mg of active material (material to be used for coating) and corresponding 5 wt% PVDF (as a binder) were thoroughly ground, and then 4–8 drops of NMP were added to engulf and wet the entire powder

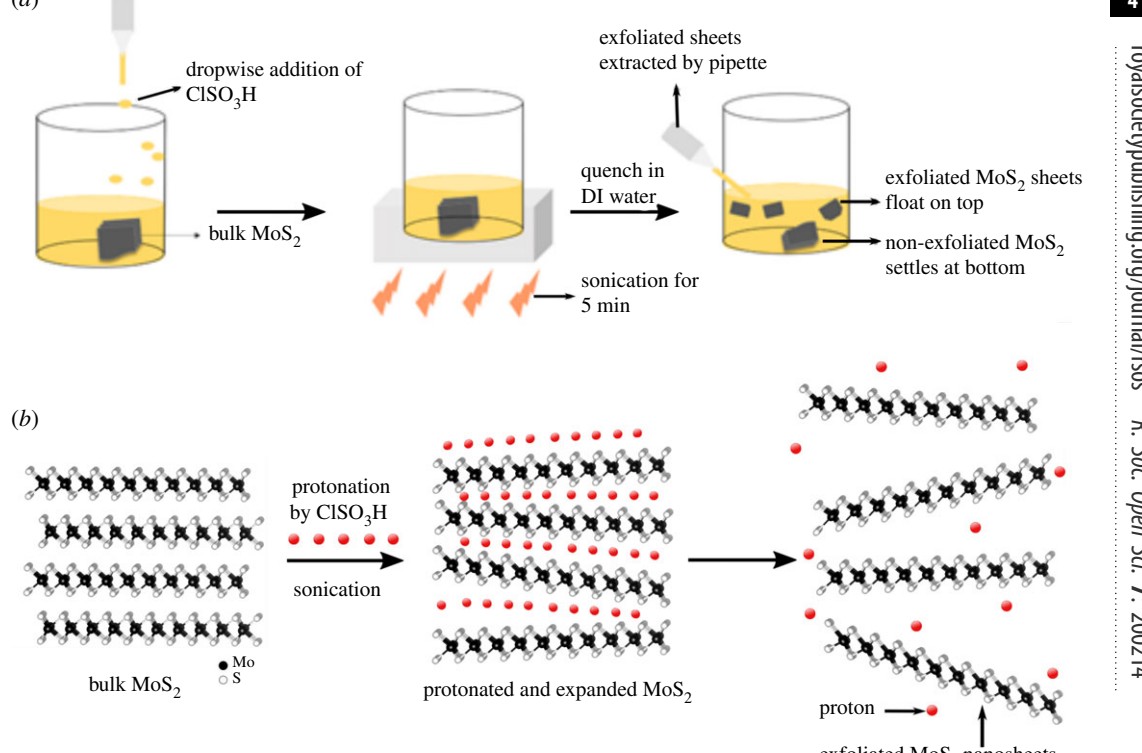

**Figure 2.** Superacid-based exfoliation process to obtain $MoS_2$ nanosheets as described in [33,34]. (*a*) Dropwise addition of $ClSO_3H$ acid to bulk $MoS_2$ followed by sonication and extraction of exfoliated sheets from the top of the solution via pipette. (*b*) Expected mechanism by which the bulk $MoS_2$ undergoes exfoliation based on [33,34].

mixture. Further thorough grinding resulted in the formation of a thick slurry (honey-like consistency) that was subsequently sonicated for approximately 10 min in isopropanol to improve its uniformity. The slurry was then pasted with a 5/32″ flat paint brush on pre-treated (in 4 M NaCl for 5 days to initiate the corrosion process) SS substrates. The pre-treated non-coated SS sample is hereafter referred to as the 'bare' sample.

The slurry-coated SS substrates were dried in an oven at 55°C for 24 h for subsequent imaging and electrochemical testing. Regarding homogenization of the coating thicknesses, identical weights of active material, binder and solvent were used for slurry preparation, and almost the same volume was painted by use of paint brush on various SS substrates. The exposed area of the SS pendulums was 2 cm$^2$. Images at every stage of the process are shown in the form of a flow diagram in figure 3.

## 2.3. Testing and analysis

X-ray diffraction (XRD) analyses were performed with a Bruker D8™ diffractometer (25°–65°). Surface morphologies of the bare and coated SS samples were observed using a Carl Zeiss™ EVO MA 10 scanning electron microscope (SEM). Energy dispersive spectra (EDS) were collected using a Zeiss Gemini SEM at 10–30 keV. To observe layered electrode morphology, a focused $Ga^+$ ion beam (Zeiss Auriga FIB-SEM) was used.

For electrochemical/corrosion testing purposes, a three-electrode set-up was used, and the electrolyte was 3.5 wt% aqueous NaCl solution. The working electrode was the SS (bare and coated) with an approximate exposed surface area of 2 cm$^2$, the reference and the counter electrode in the set-up were Ag/AgCl and Pt wire, respectively. This set-up was connected to a CHI 660E electrochemical workstation (CH Instruments, Inc.™) for the testing to be performed (at room temperature).

The OCP measurements were recorded for a duration of 60 min to attain the potential at system equilibrium. Potentiodynamic plots were obtained by performing the scan at the corresponding open circuit potentials at a rate of 0.125 mV s$^{-1}$. Electrochemical impedance spectroscopy (EIS) was measured over 1 Hz and 1000 kHz frequencies periodically by applying a sinusoidal voltage of

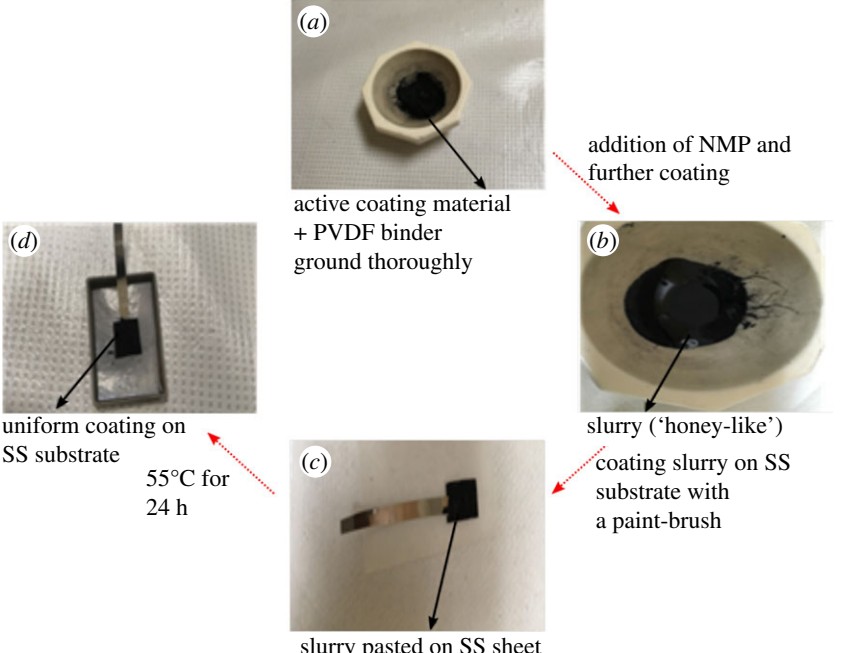

**Figure 3.** Step-by-step process of coating preparation. (*a*) Thorough grinding of the active material with PVDF to obtain a very fine powdery mixture. (*b*) Further grinding with the addition of NMP droplets to produce the slurry paste with required 'honey-like' consistency. (*c*) Pasting the slurry on the SS substrates with a paint brush. (*d*) Uniform coating obtained after drying the SS substrate in an oven at 55°C for 24 h.

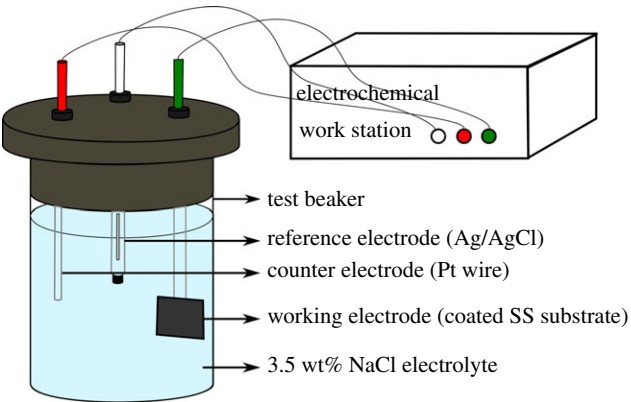

**Figure 4.** Electrochemical analysis three-electrode set-up. Schematic of the three-electrode set-up, with the counter, working and the reference electrodes and their characteristic colour codes connected to the workstation.

amplitude 5 mV. The schematic of the three-electrode set-up used for performing the experiments is shown in figure 4.

## 3. Results and discussion

Figure 5 shows SEM micrographs of the coatings before corrosion. SEM micrograph of the graphite illustrates the characteristic multi-layered planar structure (figure 5*a*), while exfoliated $MoS_2$ demonstrated stacked, thin $MoS_2$ layers (figure 5*b*). Exfoliated BN flakes (figure 5*c*) were much thinner and smaller, resulting in a coated surface that looked relatively smooth.

Figure 5*d,e* shows elemental mappings of the cross-sectional images of the graphite and exfoliated $MoS_2$ coatings, respectively. Figure 5*d* shows the distribution of all elements in the graphite coating. Elemental mapping of the coating revealed a dense carbon network throughout the coating. Exfoliated $MoS_2$ demonstrated a concentrated distribution of Mo and S network (figure 5*e*). Other elements

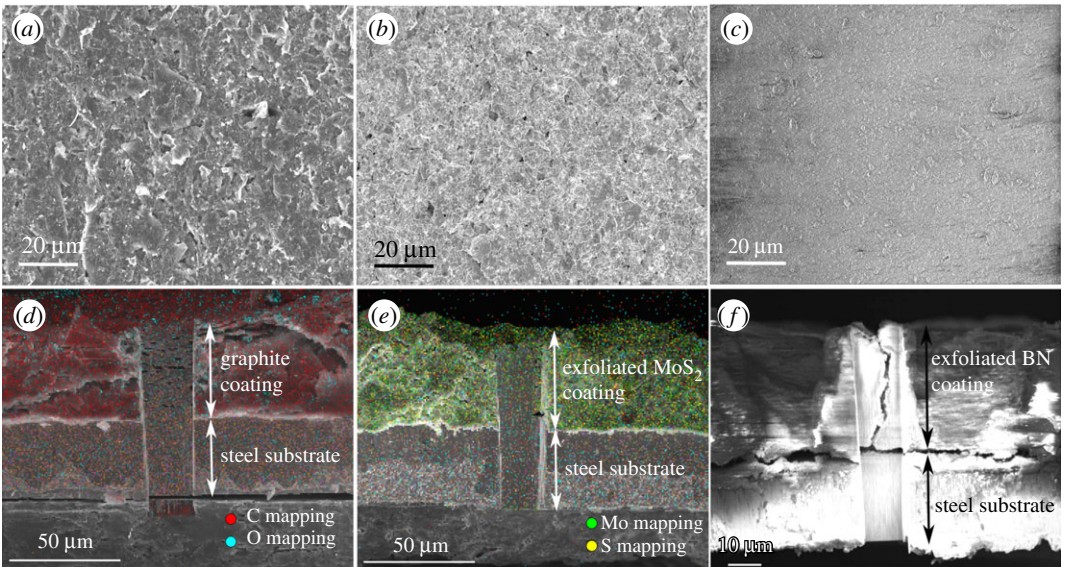

**Figure 5.** SEM micrographs and elemental mappings of coating surfaces before exposure to corrosive environment. (*a–c*) SEM micrographs of the morphologies of graphite, MoS₂ and BN coatings, respectively, before being exposed to the corrosive NaCl environment. The flaky nature of graphite coating, thin-layered structure of MoS₂ coating and relatively uniform morphology of the BN coatings are observed. (*d–f*) Cross-sectional images of various coatings, with thicknesses of approximately 50 μm for graphite and exfoliated MoS₂ coatings and approximately 60 μm for exfoliated BN coating.

found in the elemental mappings were steel elements (Fe, Mn and Cr). Figure 5*f* shows a FIB image of the cross-section of the exfoliated BN, since it could not be detected by EDS. However, the cross-sectional images were used to measure the coating thicknesses, which were approximately 50 μm for graphite and exfoliated MoS₂ coatings and approximately 60 μm for exfoliated BN coating. The vertical feature in the middle of the SEM micrographs (figure 5*d–f*) is the cut/section made by the ion beam during milling in the FIB microscope.

Figure 6 shows XRD of the exfoliated samples. XRD spectra of the exfoliated (MoS₂ and BN) and bulk samples (graphite) demonstrated characteristic peaks; the indexed peaks were matched with the Inorganic Crystal Structure Database (ICSD) and with the literature [36–39]. The peaks corresponded to hexagonal and trigonal prismatic structures for BN and MoS₂, respectively (shown in the inset).

OCP results of the bare and the coated SS samples are shown in figure 7. OCP analysis was a preliminary investigation of the corrosion resistance of coatings without any applied potential. All the materials showed a positive trend of potentials, indicating corrosion resistance due to inhibited penetration of corrosive ions on the alloy substrate [40,41]. OCP results for the MoS₂-coated sample were the most positive (approx. 0.133 V) compared to the other samples, thereby proving its superiority, followed by the BN sample (approx. 0.06 V). The graphite coating performed slightly better than the bare SS, implying that graphite coating can reduce corrosion in NaCl solution. The result for PVDF is shown in the electronic supplementary material, figure S1.

Potentiodynamic plots have been used to study the pitting corrosion-resistant properties of the coatings and the bare-steel sample under an applied potential and have more significance because these more accurately mimic the pitting corrosion mechanism to a greater extent than simple OCP analysis [42]. The potentiodynamic plots of the samples are shown in figure 8.

In this case, the potential scan was performed at the corresponding open circuit potential to accurately observe the corrosion potential ($E_{corr}$) and current ($I_{corr}$) values. Using a three-electrode set-up is advantageous because it closely identifies the potentials with respect to the working electrode (in this case, the coated and the bare SS) and the counter [43]. From figure 8 and table 1, it is observed that the MoS₂-coated sample demonstrated corrosion current density of $I_{corr}$ (0.05 μA cm⁻²) and the most positive $E_{corr}$ value (0.08 V) of the samples, again indicating its strong anti-corrosive properties under the applied potential. Although BN coating showed corrosion current density value of $I_{corr}$ (0.05 μA cm⁻²), the corrosion potential $E_{corr}$ (−0.22 V) shifted towards the negative side of the potentiodynamic polarization graph. Graphite coating showed the highest corrosion rate (0.0036 mm yr⁻¹), indicating its tendency to promote corrosion due to its electrical conductivity.

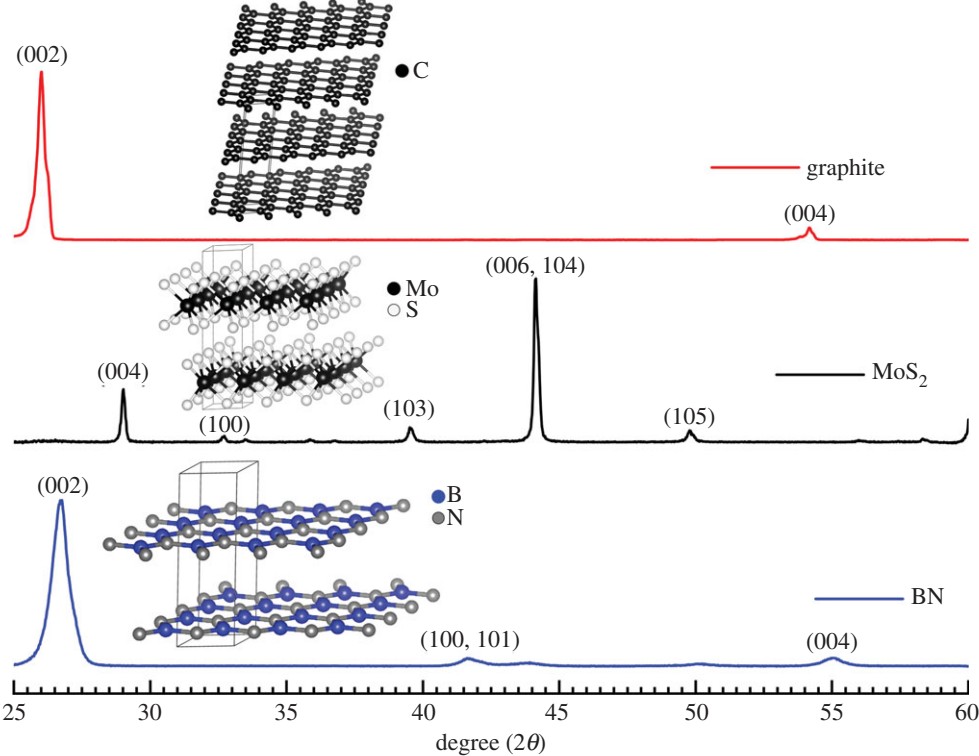

**Figure 6.** XRD spectra of layered 2D materials. XRD spectra stack of the exfoliated BN, MoS$_2$ (blue and black, respectively) and bulk as-obtained graphite (red) showing the characteristic peaks. Respective unit cells are shown in the inset.

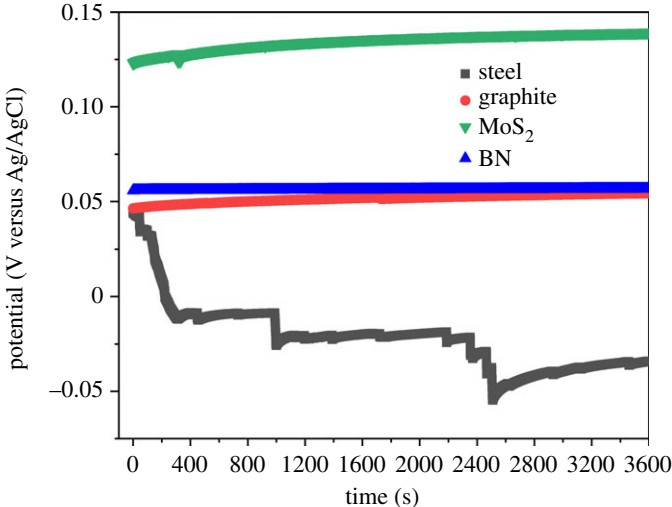

**Figure 7.** OCP plots indicating coated and bare SS samples taken over a period of 60 min in 3.5% NaCl solution. The exfoliated materials (MoS$_2$ and BN) demonstrate higher OCVs than graphite coating and the bare steel sample.

Table 1 provides quantitative values of the corrosion rate (CR), demonstrating that MoS$_2$ provided the best coating efficiency while bare SS fared better than graphite coatings in NaCl media.

CR values in table 1 were obtained using equation (3.1),

$$CR = \left( K.\frac{I_{corr}}{\rho.A} \right) * E_W,  \tag{3.1}$$

where $K$ is the CR constant with a value of $3.27 \times 10^{-3}$ mm g ($\mu$A cm year)$^{-1}$, $I_{corr}$ is the corrosion current ($\mu$A), $\rho$ is the material density (7.87 g cm$^{-3}$), $E_W$ is the equivalent weight of the material (27.56 g) and $A$ is the area of active material on the electrode exposed to the corrosive media (2 cm$^2$) [40]. Results in table 1

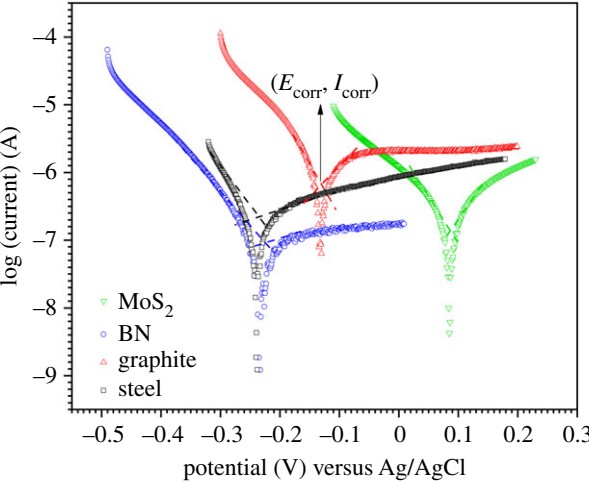

**Figure 8.** Anodic and cathodic polarization curves of coated and bare SS samples at the corresponding OCP at a scan rate of 0.125 mV s$^{-1}$ and the corresponding corrosion currents and voltages (the point of crossover of the dotted lines for each sample).

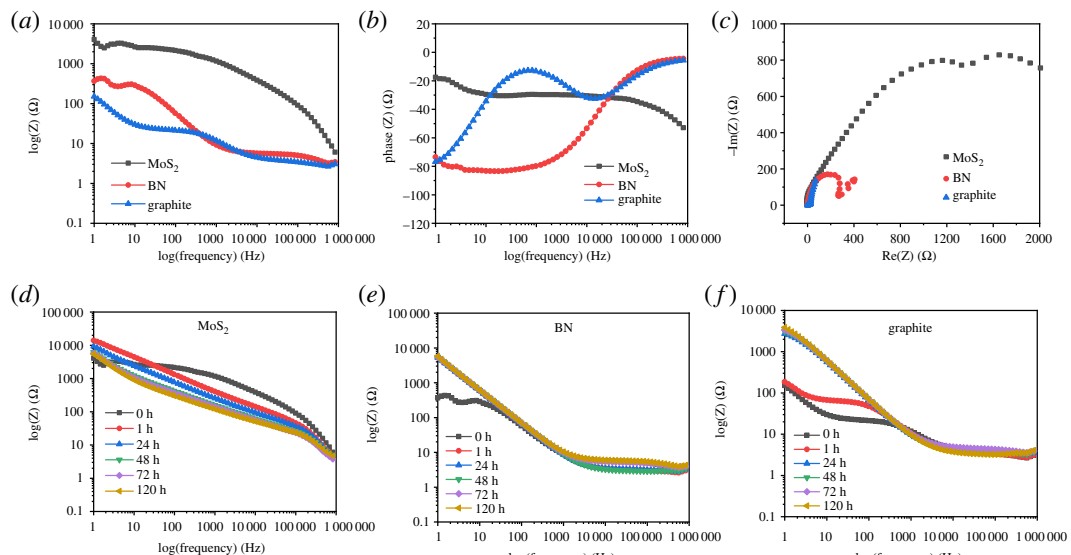

**Figure 9.** EIS test results of exfoliated MoS$_2$, BN and bulk graphite-coated steel in 3.5 wt% NaCl solution. (a–c) Bode and Nyquist plots of MoS$_2$, BN and graphite at the onset of exposure (t = 0 h). Impedance spectra of, (d–f) MoS$_2$, BN and graphite coatings over long exposure time in corrosive environment.

**Table 1.** Corrosion rate calculation of various coatings and bare SS sample. CR, corrosion rate.

| tested sample | $E_{corr}$ (V) | $I_{corr}$ (μA cm$^{-2}$) | CR (mm yr$^{-1}$) |
|---|---|---|---|
| MoS$_2$ coated | 0.08 | 0.05 | 0.0005 |
| BN coated | −0.22 | 0.05 | 0.0005 |
| bare | −0.24 | 0.155 | 0.0017 |
| graphite coated | −0.13 | 0.315 | 0.0036 |

indicate that bulk graphite may not be a suitable anti-corrosion coating in aqueous saline media. In addition, the low corrosion rate values of bare SS were potentially due to the formation of a passivating layer on its surface that prevents further corrosion [44].

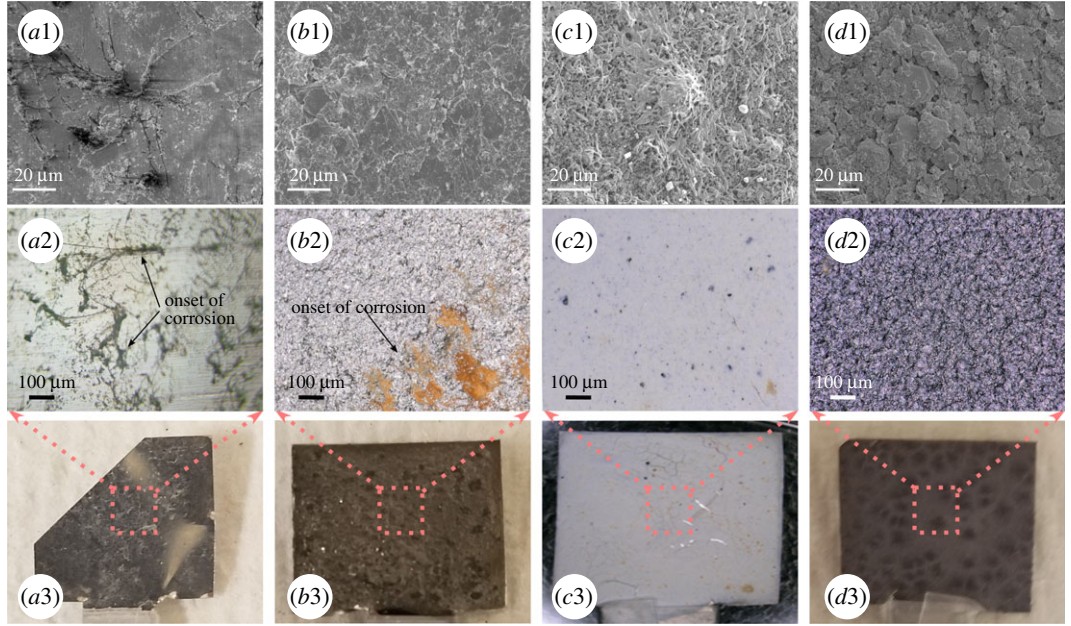

**Figure 10.** SEM, optical and digital images of the (*a*) steel, (*b*) graphite, (*c*) BN and (*d*) MoS$_2$-coated surfaces after exposure to a corrosive NaCl environment. Corrosion on the bare SS, flaky and corrosive nature of graphite coating, nanotube morphology of the BN coating and a stack of thin platelets in MoS$_2$ coatings are observed.

EIS was carried out periodically by applying a sinusoidal voltage of amplitude 5 mV; responses were measured over 1 Hz and 1000 kHz frequencies. Figure 9 compares exfoliated MoS$_2$, BN and graphite coatings at the beginning of exposure ($t = 0$ h) with these plots depicting an order of magnitude difference in impedance values. In figure 9*a*–*c*, Nyquist and Bode plots of MoS$_2$-coated steel show higher capacitive impedance behaviour, which indicates that MoS$_2$ coating more efficiently prevented dissolved oxygen, water and chloride ions from penetrating the steel surface compared to BN and graphite. Figure 9*b* shows a higher phase angle for MoS$_2$ than the BN and graphite coatings, indicating the probability of corrosion prevention against water permeation. In figure 9*c*, the resistance semicircle for MoS$_2$ coating is visible in the high-frequency range and, largest among the coatings. By contrast, BN and graphite coatings showed relatively smaller semicircles. The larger the semicircle, the higher the charge transfer resistance [30,45]. Therefore, MoS$_2$-coated steel showed the best performance in corrosion protection, as the corrosion of steel depends on the charge transfer process. Figure 9*d*–*f* shows time-dependent EIS plots and comparison between the coatings, depicting the magnitude difference between the impedance values at varying times. As time increased, the impedance values of the coatings gradually decreased, making them susceptible to corrosion [46]. Comparison of EIS test results among the bare steel, exfoliated MoS$_2$, BN and bulk graphite-coated steel is shown in the electronic supplementary material, figure S3–S4.

Figure 10 shows SEM micrographs of the bare and coated SS samples after 240 h in 3.5 wt% NaCl, providing a vital imaging perspective to the corrosive effect of the NaCl environment.

Because the coatings were intended to provide a protective layer, a loss in the integrity of the coating will be detrimental to the corrosion resistance. Figure 10*a*1 shows the penetration of salt into the outer layer of bare SS, resulting in localized corrosion on the surface. Visible corrosion on the surface appears in dark contrast in figure 10*a*2. A trace of white salts is evident on the steel surface figure 10*a*3. SEM micrograph of graphite in figure 10*b*1 demonstrates a flaky and non-homogeneous surface, resulting in poor anti-corrosive properties. Optical image of the graphite coating in figure 10*b*2 shows typical corrosion in the bottom right corner. Figure 10*c*1 shows a densely entangled network of very small BN particles. The optical and digital images of BN coating in figure 10*c*1 and *c*2, respectively, reveal damage to the coating in some places after exposure to NaCl, allowing paths for corrosion inception on the steel substrate. For MoS$_2$ coating shown in figure 10*d*1, thin and layered small particles are evident after exposure to corrosive environment. These packed granular structures effectively inhibit corrosion. The optical image in figure 10*d*2 and digital image in figure 10*d*3 of MoS$_2$ coating displays a smooth coating without any trace of corrosion (compared to bare steel and other coatings), and only tiny particles can be observed after immersion in a corrosive

environment. Optical and digital images of the bare steel, before and after pre-treatment, are presented in the electronic supplementary material, figure S5, showing the onset of corrosion process on the steel.

# 4. Conclusion

Results showed that a uniform 2D nanomaterial-based coating is an important criterion for developing anti-corrosion protective layers. Coatings with minor breaks or gaps resulted in localized corrosion at those points. OCP, potentiodynamic plot and pitting corrosion rate calculations indicated superior anti-corrosion properties of the exfoliated $MoS_2$, which can be attributed to the coating uniformity and large flake size. In addition to coating uniformity, $MoS_2$ has been reported to have a high energy barrier to oxygen ion infiltration ($\Delta E \sim 8$–$10\,eV$), leading to superior corrosion protection properties. Long-term electrochemical test and visual inspection of the coated samples also confirmed the long-term stability of the $MoS_2$-coated SS in the corrosive medium. BN's anti-corrosion properties may be attributed to its insulating nature, which prevents galvanic corrosion of SS. The graphite-coated SS sample did not perform as well, due to lack of a uniform protective layer and the presence of minor gaps or breaks on its surface through which ions can penetrate, resulting in further propagation of corrosion.

Data accessibility. Data available from the Dryad Digital Repository: https://doi.org/10.5061/dryad.n02v6wwt5 [47]. The datasets supporting this article have been uploaded as part of the electronic supplementary material.

Authors' contributions. S.B.M. and S.M. prepared the coatings and performed testing and analysis. Z.R. contributed to the interpretation of data. G.S. conceived the idea, designed the experiments and co-wrote the manuscript with S.B.M., S.M. and Z.R. All authors discussed the results and commented on or revised the manuscript.

Competing interests. The authors declare no conflict of interest. The founding sponsors had no role in the design of the study; in the collection, analyses or interpretation of data; in the writing of the manuscript and in the decision to publish the results.

Funding. This work is supported by National Science Foundation grant no. 1454151.

Acknowledgements. We thank Diana Arreola and Monsur Abass from Kansas State University for their help with the initial experiments involving coating preparation. We also thank Drs Elisabeth Mansfield and Jason Holm from National Institute of Standards and Technology, Boulder, CO for their help with the FIB, SEM and EDS data collection and analyses.

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
