## [Reviewer comments · Royal Society Open Science]

Review History

RSOS-200214.R0 (Original submission)

Review form: Reviewer 1

Is the manuscript scientifically sound in its present form?

Yes

Are the interpretations and conclusions justified by the results?

Yes

Is the language acceptable?

Yes

Do you have any ethical concerns with this paper?

No

Have you any concerns about statistical analyses in this paper?

No

Recommendation?

Accept with minor revision (please list in comments)

Comments to the Author(s)

Dear Authors,

Help address the below concerns in the manuscript;

* Relabel images in figure 5

* Figure 10(a1,a2,b1,b2): Since non-exposed images are not shared; it may either help if the authors could share the as-prepared images or highlight the corrosion areas/features.

* Share the equipment/setup details of XRD and EIS.

* It may help to highlight a corrosion relevant frequency range in the EIS frequency sweep.

* In Fig.10, were the samples imaged (optical/SEM) as-is or had to be prepared to remove sample artifacts?

thank you.

Review form: Reviewer 2

Is the manuscript scientifically sound in its present form?

Yes

Are the interpretations and conclusions justified by the results?

Yes

Is the language acceptable?

Yes

Do you have any ethical concerns with this paper?

No

Have you any concerns about statistical analyses in this paper?

No

Recommendation?

Accept with minor revision (please list in comments)

Comments to the Author(s)

The authors have presented a through corrosion resistance analysis of stainless steel with a coating based on different prototypical 2D materials. This research work is suitable for publication in this journal. However, the manuscript requires some revisions before consideration for publication in this journal.

Comments:

1. The authors should include the thickness measurement of MoS₂ and BN to prove the efficiency of the exfoliation. What was the thickness of the graphite flakes?
2. The authors claimed that the BN coating looked much smoother due to thinner and smaller BN flakes. Could it be possible that the coating roughness is originating from the paintbrush? What kind of paintbrush was used during the coating procedure? How did the thickness control was achieved while using the paintbrush?

Decision letter (RSOS-200214.R0)

23-Mar-2020

Dear Mr Mujib,

On behalf of the Editors, I am pleased to inform you that your Manuscript RSOS-200214 entitled "Assessing Corrosion Resistance of 2D Nanomaterial-based Coatings on Stainless Steel Substrates" has been accepted for publication in Royal Society Open Science subject to minor revision in accordance with the referee suggestions. Please find the referees' comments at the end of this email.

The reviewers and handling editors have recommended publication, but also suggest some minor revisions to your manuscript. Therefore, I invite you to respond to the comments and revise your manuscript.

- Ethics statement

- Data accessibility

<http://datadryad.org/submit?journalID=RSOS&manu=RSOS-200214>

- Competing interests

- Authors' contributions

AB carried out the molecular lab work, participated in data analysis, carried out sequence alignments, participated in the design of the study and drafted the manuscript; CD carried out the statistical analyses; EF collected field data; GH conceived of the study, designed the study,

coordinated the study and helped draft the manuscript. All authors gave final approval for publication.

- Acknowledgements

- Funding statement

Because the schedule for publication is very tight, it is a condition of publication that you submit the revised version of your manuscript before 01-Apr-2020. Please note that the revision deadline will expire at 00.00am on this date. If you do not think you will be able to meet this date please let me know immediately.

If your manuscript is newly submitted and subsequently accepted for publication, you will be asked to pay the article processing charge, unless you request a waiver and this is approved by Royal Society Publishing. You can find out more about the charges at <https://royalsocietypublishing.org/rsos/charges>. Should you have any queries, please contact openscience@royalsociety.org.

on behalf of Professor Hazel Assender (Associate Editor) and R. Kerry Rowe (Subject Editor)
openscience@royalsociety.org

Associate Editor Comments to Author (Professor Hazel Assender):

The authors should address each of the minor changes suggested and make specific reference to any effect of the deposition (e.g. effect of paintbrush) on the observed roughness.

Reviewer comments to Author:

Reviewer: 1

Comments to the Author(s)

Dear Authors,

Help address the below concerns in the manuscript;

* Relabel images in figure 5

* Figure 10(a1,a2,b1,b2): Since non-exposed images are not shared; it may either help if the authors could share the as-prepared images or highlight the corrosion areas/features.

* Share the equipment/setup details of XRD and EIS.

* It may help to highlight a corrosion relevant frequency range in the EIS frequency sweep.

* In Fig.10, were the samples imaged (optical/SEM) as-is or had to be prepared to remove sample artifacts?

thank you.

Reviewer: 2

Comments to the Author(s)

The authors have presented a thorough corrosion resistance analysis of stainless steel with a coating based on different prototypical 2D materials. This research work is suitable for publication in this journal. However, the manuscript requires some revisions before consideration for publication in this journal.

Comments:

1. The authors should include the thickness measurement of MoS₂ and BN to prove the efficiency of the exfoliation. What was the thickness of the graphite flakes?
2. The authors claimed that the BN coating looked much smoother due to thinner and smaller BN flakes. Could it be possible that the coating roughness is originating from the paintbrush? What kind of paintbrush was used during the coating procedure? How did the thickness control was achieved while using the paintbrush?

Author's Response to Decision Letter for (RSOS-200214.R0)

See Appendix A.

Decision letter (RSOS-200214.R1)

01-Apr-2020

Dear Mr Mujib,

It is a pleasure to accept your manuscript entitled "Assessing Corrosion Resistance of 2D Nanomaterial-based Coatings on Stainless Steel Substrates" in its current form for publication in Royal Society Open Science. The comments of the reviewer(s) who reviewed your manuscript are included at the foot of this letter.

Due to rapid publication and an extremely tight schedule, if comments are not received, your paper may experience a delay in publication. Royal Society Open Science operates under a continuous publication model. Your article will be published straight into the next open issue and this will be the final version of the paper. As such, it can be cited immediately by other researchers. As the issue version of your paper will be the only version to be published I would

advise you to check your proofs thoroughly as changes cannot be made once the paper is published.

on behalf of Professor Hazel Assender (Associate Editor) and R. Kerry Rowe (Subject Editor)
openscience@royalsociety.org

Appendix A

Assessing Corrosion Resistance of 2D Nanomaterial-based Coatings on Stainless Steel Substrates (Manuscript id: RSOS-200214)

AUTHORS' RESPONSE TO REVIEWERS' COMMENTS

Associate Editor Comments to Author (Professor Hazel Assender):

Comment: The authors should address each of the minor changes suggested and make specific reference to any effect of the deposition (e.g. effect of paintbrush) on the observed roughness.

Response: Thank you for reviewing the manuscript. We have responded to each of the reviewers' comments and made necessary changes according to the suggestions.

Reviewer 1: Comments to the Author(s)

Dear Authors,

Help address the below concerns in the manuscript;

Comment #1: Relabel images in figure 5

Response #1: Thank you for pointing this out. The figures have been relabeled and modified.

CHANGES MADE: Modified Figure 5 is included in the revised manuscript.

Figure 5. SEM micrographs and elemental mappings of coating surfaces before exposure to corrosive environment.

Comment #2: Figure 10(a1,a2,b1,b2): Since non-exposed images are not shared; it may either help if the authors could share the as-prepared images or highlight the corrosion areas/features.

Response #2: SEM images non-exposed/as-prepared samples (i.e. coated samples before exposure to corrosive environment) are shown in Figure 5. The corrosion areas are highlighted in Figure 10.

CHANGES MADE: Modified Figure 10 is included in the revised manuscript.

Figure 10. SEM, optical and digital images of the (a) steel, (b) graphite, (c) BN, and (d) MoS₂ coated surfaces after exposure to a corrosive NaCl environment.

Comment #3: Share the equipment/setup details of XRD and EIS.

Response #3: X-ray diffraction (XRD) analyses were performed with a Bruker D8™ diffractometer in the range of 25°-65°. Samples were placed onto a sample holder assuring a flat upper surface.

Electrochemical Impedance Spectroscopy (EIS) was measured over 1 Hz and 1000 kHz frequencies periodically by applying a sinusoidal voltage of amplitude 5 mV. For electrochemical testing purposes, a 3-electrode set-up was used, and the electrolyte was 3.5 wt. % aqueous NaCl solution. The working electrode was the SS (bare and coated) with an approximate exposed surface area of 1 cm², the reference and the counter electrode in the set-up were Ag/AgCl and Pt wire, respectively. This set-up was connected to a CHI 660E electrochemical workstation (CH Instruments, Inc™) for the testing to be performed (at room temperature). The schematic of the 3-electrode setup used for performing the experiments is shown in Figure 4.

CHANGES MADE: Equipment and setup details of XRD and EIS are included in 'Testing and Analysis' part of the revised manuscript (P5).

Figure 4. Electrochemical analysis 3-electrode set-up

Comment #4: It may help to highlight a corrosion relevant frequency range in the EIS frequency sweep.

Response #4: Thank you for the comment. The high-frequency region semicircle of the EIS plot is related to the charge transfer resistance (R_{ct}), and substrate corrosion depends on the charge transfer process [1].

For example, Figure 9(c) shows the Nyquist plot of the coated samples. Here, the curve appears semicircular in the high-frequency region, and the Warburg impedance in the low-frequency region disappears for MoS₂ coated sample. This indicates that the nanomaterial coating prevents the diffusion of dissolved oxygen, water, and chloride ions from penetrating the steel substrate.

CHANGES MADE: The explanation of the corrosion relevant frequency range in the EIS plots are included in the revised manuscript (P10L262).

Figure 9. EIS test results of the exfoliated MoS₂, BN, and bulk graphite coated steel in 3.5 wt.% NaCl Solution.

Comment #5: In Fig.10, were the samples imaged (optical/SEM) as-is or had to be prepared to remove sample artifacts?

Response #5: The sample images were as-is (i.e., after exposure to the corrosive environment) in Figure 10. No additional processing was done. Samples before exposure to corrosive environment are shown in Figure 5.

CHANGES MADE: None.

Reviewer 2: Comments to the Author(s)

The authors have presented a through corrosion resistance analysis of stainless steel with a coating based on different prototypical 2D materials. This research work is suitable for publication in this journal. However, the manuscript requires some revisions before consideration for publication in this journal. Comments:

Comment #1: The authors should include the thickness measurement of MoS₂ and BN to prove the efficiency of the exfoliation. What was the thickness of the graphite flakes?

Response #1: Based on the SEM images, the average lateral size of MoS₂ particles was approx. 2 μm, and the average lateral size of BN was approx. 1 μm. Based on previous experience, we have observed that the exfoliated 2D materials mix readily with PVDF and NMP, thus forming a uniform and stable coating, which is not necessarily the case with the bulk crystals. From the previous experiments done by our group, superacid exfoliated MoS₂ was observed to be a few layers (10 layers) thick with flake size ranging from 100 nm to 1 μm [2].

Graphite flakes range in size from 50-800 μm in diameter and 1-150 μm thick according to manufacturer specifications (Sigma Aldrich™).

CHANGES MADE: The average lateral size of the MoS_2 , BN, and Graphite flakes are included in the 'Material and Methods' section of the manuscript (P3).

Comment #2: The authors claimed that the BN coating looked much smoother due to thinner and smaller BN flakes. Could it be possible that the coating roughness is originating from the paintbrush? What kind of paintbrush was used during the coating procedure? How did the thickness control was achieved while using the paintbrush?

Response #2: Yes, the coating roughness may be originating from the paint-brush. However, to ensure consistency, all the coatings were prepared with thickness of approx. (50-60) μm , as shown in Figure 5. This thickness was achieved using a 5/32" flat paint-brush, and the same coating protocol was allied for all samples. An equal amount of active material (30 mg) and binder (5 wt.%) were initially used for slurry preparation, and almost the same amount was painted on the SS substrate.

CHANGES MADE: The following lines are included in the 'Preparation of Slurry and Coatings' section of the revised manuscript (P4).

"The slurry was then pasted with a 5/32" flat paintbrush on pre-treated (in 4 M NaCl for 5 days to initiate the corrosion process) SS substrates.

Regarding homogenization of the coating thicknesses, identical weights of active material, binder, and solvent were used for slurry preparation, and almost the same volume was painted by use of paint-brush on various SS substrates."

References

- [1] Qu, Z.; Wang, L.; Tang, H.; Ye, H.; Li, M. Effect of Nano-SnS and Nano-MoS₂ on the Corrosion Protection Performance of the Polyvinylbutyral and Zinc-Rich Polyvinylbutyral Coatings. *Nanomaterials* **2019**, *9*, 956.
- [2] David, L.; Bhandavat, R.; Singh, G. MoS₂/graphene composite paper for sodium-ion battery electrodes. *ACS Nano*. **2014**, *8*, 1759-1770.